# Sexting Behaviors and Fear of Missing out Among Young Adults

**DOI:** 10.3390/bs15040454

**Published:** 2025-04-01

**Authors:** Mara Morelli, Alessandra Ragona, Antonio Chirumbolo, Maria Rosaria Nappa, Alessandra Babore, Carmen Trumello, Gaetano Maria Sciabica, Elena Cattelino

**Affiliations:** 1Department of Dynamic and Clinical Psychology, and Health Studies, Sapienza University of Rome, 00185 Rome, Italy; mara.morelli@uniroma1.it (M.M.); gaetanomaria.sciabica@uniroma1.it (G.M.S.); 2Department of Developmental and Social Psychology, Sapienza University of Rome, 00185 Rome, Italy; alessandra.ragona@uniroma1.it; 3Department of Psychology, Sapienza University of Rome, 00185 Rome, Italy; antonio.chirumbolo@uniroma1.it; 4Department of Systems Medicine, Tor Vergata University of Rome, 00133 Rome, Italy; maria.rosaria.nappa@uniroma2.it; 5Department of Psychology, University ‘G. d’Annunzio’ of Chieti-Pescara, 66013 Chieti, Italy; a.babore@unich.it (A.B.); c.trumello@unich.it (C.T.); 6Department of Human and Social Science, University of Valle d’Aosta, 11100 Aosta, Italy

**Keywords:** sexting, FoMO, young adults, risky sexting, LGB

## Abstract

Fear of missing out (FoMO) creates a strong urge to stay continuously connected and informed about peers’ activities, identified as a risk factor for problematic social media use and risky behaviors. Sexting is generally defined as the exchange of sexually suggestive or explicit photos, videos, or text messages through cell phones or other technologies. Despite its social relevance, the link between FoMO and sexting remains underexplored. This study examines their relationship in young adults—an understudied group compared to adolescents—while controlling for age, sex, and sexual orientation. The study surveyed 911 Italian young adults (18–30 years, M_age_ = 22.3, SD_age_ = 2.57, 74% women, 70.4% heterosexual) through an online questionnaire. The results indicate that FoMO predicts only risky sexting behaviors (sexting under substance use and sexting for emotion regulation) while not influencing experimental sexting (sending one’s own sexts). Additionally, the link between FoMO and sexting for emotion regulation is stronger among LGB individuals. Therefore, FoMO has proven to be strongly related to the two kinds of risky sexting but not to experimental sexting. Understanding this relationship can inform prevention and intervention programs on relationships, online communication, and sexting in young adults.

## 1. Introduction

Over the past few years, the widespread use of technology for social interactions has undoubtedly brought numerous benefits, such as bridging the gap between individuals and enabling connections with people from different parts of the world. In particular, the use of social media for social, romantic, and sexual interactions has grown significantly ([10]; [33]). However, the literature consistently emphasizes the need to address a range of inherent risks tied to technology and social media usage. Among these risks are social media addiction ([2]) and, more broadly, a decline in the well-being of young people who spend extended hours on social networking platforms ([55]).

One area of growing concern in this context is the phenomenon of fear of missing out (FoMO), which has become increasingly prevalent alongside the rise of social media ([21]). FoMO refers to a common anxiety stemming from the belief that others may be engaging in fulfilling or enjoyable experiences from which one is excluded. This feeling is often accompanied by a persistent urge to remain connected and updated on the actions and lives of others, particularly close friends, to avoid feelings of disconnection or exclusion ([54]).

Drawing on the Self-Determination Theory framework ([17]), [54] ([54]) demonstrated that individuals with lower satisfaction of core psychological needs—such as autonomy, competence, and relatedness—are more likely to experience FoMO. Researchers have further linked FoMO to feelings of personal insecurity and inadequacy, which heighten fears of social exclusion ([29]). As a result, FoMO can drive individuals to seek constant social connection.

Social media platforms, which provide continuous updates on others’ lives and activities, have been identified as a key medium through which FoMO manifests. A recent meta-analysis established a clear linear connection between FoMO and social media use ([74]), and numerous studies have also recognized FoMO as a significant factor contributing to excessive, compulsive, or problematic engagement with social media ([32]; [63]). Moreover, FoMO has been examined in relation to various negative online behaviors, including cyberbullying ([27]), problematic use of internet and social networks ([5]; [29]), and harmful patterns of smartphone and social media communication ([58]).

### 1.1. Sexting Types and Behaviors

An online behavior that has been extensively studied and remains highly controversial in recent decades is sexting, defined as the engagement in exchanging provocative or sexually suggestive messages, photos, and videos via smartphones, the internet, or social networks ([13]). The literature appears to be divided on many aspects of sexting, ranging from its definitions to the various types of behaviors associated with it ([18]; [45]). Recent meta-analyses have explored various types of sexting behaviors ([49], [48]), positioning them along a spectrum that ranges from safer to more risky actions. At the safer end of this spectrum, we find experimental sexting, which refers to a consensual exchange of explicit messages that allow individuals to explore and address developmental needs related to their sexuality and identity formation ([7]; [38]; [46]). This concept was initially proposed by [72] ([72]), who conceived sexting as a new sexual norm, particularly during adolescence and young adulthood. These developmental stages are marked by individuals’ exploration of their sexual identity and their perceptions of body image ([9]). Experimental sexting is often regarded as a means of fostering passion, intimacy, and enjoyment ([19]; [67]), particularly in long-distance relationships ([3]; [52]; [70]), and is more prevalent among sexual minorities in the context of increased communication ([14]). Several studies have also emphasized its positive effects, such as boosting self-esteem and enhancing body image ([8]). Additionally, experimental sexting can serve sexual purposes, including flirting ([1]), maintaining a relationship ([69]), or initiating sexual encounters ([64]).

On the other hand, sexting is often viewed as an aggressive, risky, and dangerous behavior, particularly when it involves minors, nonconsensual exchanges, or strangers, posing threats to privacy, safety, and mental well-being ([41], [45]; [59]). It can also have negative consequences ([68]) and may be driven by various motives, such as financial gain ([41]) or coercion ([15]). For instance, [9] ([9]) found that sexting for instrumental reasons related to secondary purposes, including seeking favors or money, as well as sexting with harmful intent, increases the risk of both perpetrating and experiencing dating violence. At the extreme end of the spectrum, aggravated sexting—characterized by harmful intentions, nonconsensual sharing of explicit content, and coercion—can lead to severe consequences such as bullying, cyberbullying, and traditional and cyberdating violence ([6]; [19]; [26]; [23], [24]; [41], [42]).

Another type of sexting identified by scholars is risky sexting ([46], [45]), which involves behaviors that, while not coercive in nature, co-occur with other risky activities. This form of sexting typically occurs when individuals engage in sexting while under the influence of substances like drugs or alcohol, or when sexting is directed towards strangers met online. A systematic review pointed to a strong association between sexting and risky behaviors ([48]), and some studies have specifically found correlations between sexting and substance use, including alcohol, ecstasy, marijuana, and cocaine ([4]). Moreover, another type of sexting considered risky is sexting for emotional regulation ([7]), in which sexting is used as a tool to cope with difficulties in emotional regulation ([16]) and to manage a low sense of emotional self-efficacy, as well as negative emotions such as anger, loneliness, and attention seeking ([73]).

Sexting behaviors are more commonly reported among adolescents and young adults, with studies showing a higher prevalence in these age groups ([36]; [39]; [49]; [60]). This age group is particularly active in exploring digital communication and sexual expression, making sexting more widespread in their social interactions. The current literature highlights age-related differences in sexting behaviors, demonstrating that young adults engage in sexting more frequently than adolescents ([7]; [39]). Evidence suggests that the sexting prevalence peaks during young adulthood ([49]), likely because individuals in this age group are still exploring their sexuality and relationships, albeit with greater awareness and maturity compared to adolescents ([47]).

### 1.2. The Roles of Age, Gender, and Sexual Orientation in Impacting Sexting Behaviors

Young adults appear more inclined to experiment with various sexual behaviors, including sexting, both within stable relationships, with a prevalence of 60%, and with casual partners or strangers, with a prevalence of 44% ([11]; [20]; [49]). Among young adults, the prevalence of sexting behaviors varies widely. [49] ([49]) found that the percentage of individuals sending sexts ranges from 32% to 44.6%, while receiving sexts occurs in 31.9% to 51.2% of cases. Both experimental and risky sexting appears to become more common as individuals grow older, as highlighted in a cross-cultural study by [43] ([43]).

The evidence on gender differences in sexting remains inconclusive. Some research suggests that females engage in generic sexting behaviors more frequently ([69]), while other studies report higher rates among males ([31]). In contrast, a meta-analysis by Madigan and colleagues (2018) observed no significant gender differences. More specifically, [43] ([43]) noted that young adult males are more likely than females to engage in aggravated and risky sexting behaviors, and in general, males tend to be more inclined to engage in high-risk activities ([37]).

Sexting patterns also vary based on sexual orientation. Sexual minorities appear more likely to participate in sexting ([7]; [43]; [67]), and in general, are at a heightened risk of engaging in risky sexual behaviors ([50]), possibly because online spaces offer a sense of protection from societal stigma and prejudice. This aligns with [40]’s ([40]) Minority Stress Model, which suggests that virtual environments may reduce stressors linked to discrimination ([14]).

### 1.3. Fear of Missing Out (FoMO) and Sexting

The relationship between FoMO and sexting has been explored in only a limited number of studies, producing mixed findings ([44]). Some research identifies FoMO as a significant predictor of erotic sexting ([53]; [61]), while other studies report no connection ([66]). The recent study by [44] ([44]) highlighted that FOMO predicts three motivations for sexting: sexual purposes, body image reinforcement, and instrumental/aggravated motivations. It emphasizes the importance of further studying this factor in determining potentially risky sexting behaviors. Age seems to play a crucial role in shaping both FoMO and sexting behaviors, albeit in different ways. When it comes to FoMO, younger individuals are generally more vulnerable to its effects, as highlighted by [54] ([54]). On the other hand, sexting behaviors show a different trend. Older individuals are often more motivated to engage in sexting for sexual purposes compared to younger age groups ([9]). The only study that investigated the effect of age on the relationship between sexting and FoMO ([44]) did not find any direct or interaction effect, despite previous studies showing that involvement in sexting increases with age ([9]), both among adolescents and young adults. Conversely, the study by [44] ([44]) demonstrated the moderating role of sexual orientation in sexting motivations, showing that LGB individuals are more likely than heterosexual ones to engage in sexting for both sexual and non-sexual reasons ([28]). Moreover, sexual orientation moderated the relationship between FoMO and sexting for body image reinforcement or instrumental/aggravated motives, with this association being stronger among LGB individuals.

Due to the scarce findings currently available in the literature, investigating the nuances of online communication, particularly its potentially problematic dimensions, is vital for shedding light on the social dynamics that may contribute to risky behaviors such as sexting. Those behaviors are not solely driven by sexual intent but often arise from a complex interplay of psychological, relational, and situational factors. Understanding these underlying mechanisms could provide valuable insights into the broader motivations and circumstances that foster these behaviors in digital contexts.

### 1.4. Aims of the Present Study

Building on the limited research examining the relationship between FoMO and sexing (e.g., [44]), the Self-Determination Theory ([17]; [54]) provides a useful framework for understanding this dynamic. According to this theory, FoMO may fuel an intense drive to establish and maintain social connections as a way to fulfill a sense of belonging. This perspective suggests a reciprocal relationship between FoMO and the motivations underlying the satisfaction of psychological and relational needs. Sexting, as a form of online communication, offers a means to address relational needs tied to sexual and erotic expression, potentially heightened by the compulsion to remain socially connected. Consequently, individuals with elevated levels of FoMO may feel a stronger motivation to engage in sexting behaviors ([61]).

The relationship between FoMO and sexting has been explored in only a few studies ([44]; [66]), and the one by [44] ([44]) focused on sexting motivations, not on different kinds of sexting behaviors. Moreover, to our knowledge, no research has yet examined the connection between FoMO and sexting behaviors in young adults or considered how factors such as age, biological sex, and sexual orientation might influence this relationship. For those reasons, the present study aimed to fill existing gaps in the literature by exploring the relationship between FoMO and sexting behaviors among Italian young adults. Specifically, the study sought to investigate how FoMO relates to distinct types of sexting behaviors (experimental sexting, i.e., sharing own sexts, and two kinds of risky sexting, i.e., sexting for emotional regulation and sexting under substance use or with strangers), while accounting for the influence of age, biological sex, and sexual orientation, as previous research has shown these factors can impact sexting behaviors ([9]).

Specifically, building on previous research (2024), we hypothesized the following:

**H1.** 
*FoMO would predict both experimental and risky sexting behaviors.*


**H2.** 
*Age can moderate the relationship between FoMO and sexting behaviors, expecting it to be stronger among older individuals compared to younger ones ([9]).*


**H3.** 
*The relationship between FoMO and risky sexting will be stronger in males. Despite conflicting findings in the literature regarding gender differences in sexting behaviors, several studies ([37]; [43]) suggest that males may exhibit greater involvement in the two risky types of sexting: sexting under the influence of substances or with strangers, and sexting for emotional regulation.*


**H4.** 
*The relationship between FoMO and each type of sexting behavior would be stronger among LGB+ individuals compared to heterosexual individuals ([9]; [25]; [35]; [44]; [51]).*


## 2. Materials and Methods

### 2.1. Participants and Procedure

This study was conducted on a sample of 911 Italian young adults (M_age_ = 22.3; SD_age_ = 2.57; age range = 18–30 years; 74% women), including both heterosexual and LGB individuals (70.4% heterosexual; *n* = 641), recruited from various Italian universities. In terms of family socio-economic status, 23% of participants reported a low income, while 77% indicated a medium to high income. As for relationship status, 35.7% of participants reported having been in a relationship in the past, while the majority (64.3%) reported being in a relationship at the time of completing the questionnaire.

Young adult participants were recruited from various Italian universities through the distribution of an anonymous link containing the research questionnaires. Participants were also asked to share the link to the survey among their contacts and on their social network platforms. This facilitated a snowball sampling approach as the survey was further shared online. The study was presented to participants as research on relationships in the digital context, and participation was entirely voluntary. A potential bias stems from the voluntary nature of participation and the non-probabilistic sampling method, which is common in most psychological studies. Before completing the online survey, participants provided their consent to participate by selecting “Yes, I accept to participate in the study” on the survey’s first page. This study adhered to the principles outlined in the Declaration of Helsinki and received approval from the university’s ethical committee of the Department of Dynamic and Clinical Psychology, and Health Studies of Sapienza University of Rome.

### 2.2. Measures

Demographic Information. Participants provided information about their biological sex, age, sexual orientation, and dating relationship status. The Kinsey scale ([34]) was used to measure sexual orientation, with participants rating themselves on a 7-point scale, where a score of 1 indicated exclusively heterosexual and a score of 7 signified not exclusively heterosexual. In accordance with the procedure used in previous studies (e.g., [44]), participants were divided into two categories: those identified as exclusively heterosexual were assigned a score of 0, while those belonging to sexual minorities received a score of 1.

Fear of Missing Out. Fear of missing out was assessed by the Fear of Missing Out Scale (FoMOs) ([54]; Italian validation by [12]). This scale is composed by 10 items rated on a 5-point Likert scale of 1 (not at all true for me) to 5 (extremely true for me). The FoMO scale explores feelings of anxiety, fear, worry, and irritability linked to the concept of missing out on rewarding experiences and the desire to maintain connections with others (a sample item is “I get worried when I find out that my friends are having fun without me”). The scale showed good reliability, with a McDonald’s ω of 0.88.

Sexting Behaviors. Sexting behaviors were assessed using 10 items from the Sexting Behavior Questionnaire (SBQ; [41]). For the purpose of this research, we only considered the following sexting dimensions selected in previous studies ([7]; [42]): experimental sexting (i.e., sharing own sexts, 3 items; a sample item is “How often have you publicly sent provocative or sexually suggestive videos about yourself?”; McDonald’s ω 0.82), and two types of risky sexting, (a) sexting during substance use or with strangers met online (4 items; a sample item is “How often do you send a sext when you smoke marijuana?”; McDonald’s ω 0.64) and (b) sexting for emotional regulation (3 items; a sample item is “How often do you send a sext when you feel lonely?”; McDonald’s ω 0.84). Participants reported the frequency of the engagement in each sexting behavior over the past 12 months, using a 5-point Likert scale from 1 (never) 5 (always or almost daily).

### 2.3. Data Analysis

Sexting dimensions showed a strongly polarized non-normal distribution. Therefore, we conducted negative binomial regression analyses in which each dimension of sexting (i.e., experimental, risky during substance use or with strangers, and sexting for emotional regulation) was regressed on age, biological sex (0 = male; 1 = female), sexual orientation (0 = exclusively heterosexual; 1 = exclusively non-heterosexual), FoMO, and the three interaction terms between FoMO and each investigated socio-demographic variable (i.e., FoMO*age, FoMO*biological sex, FoMO*sexual orientation). Negative binomial regression is a statistical technique used to model count data, representing the number of times an event occurs within a given period or space (e.g., behavioral frequencies, event counts). It is particularly useful for behavioral frequency data, including self-reported measures, which often deviate from a normal distribution. Common examples include the number of problematic behaviors, the frequency of specific actions performed by an individual, or other self-reported behavioral measures in psychological and social research ([30]). Version 2.5.5 of jamovi ([65]) with the GAMLj3 module ([22]) was used for the analyses.

## 3. Results

### Negative Binomial Regression

The negative binomial regression model for experimental sexting, which is the sharing of own sexts, accounted for 6% of the variance, R^2^ = 0.06, χ^2^(7) = 24.6, *p* < 0.001. Experimental sexting demonstrated a positive association exclusively with sexual orientation, B = 0.16, SE = 0.03, Exp(B) = 1.17, *p* < 0.001. See Table 1 for the statistics.

Regarding the first type of risky sexting (sexting under substance use or with strangers), the negative binomial regression model accounted for 8.5% of the variance, R^2^ = 0.085, χ^2^(7) = 40.3, *p* < 0.001. Specifically, this kind of risky sexting was found to be positively and significantly related to sexual orientation, B = 0.17, SE = 0.03, Exp(B) = 1.19, *p* < 0.001, and FoMO, B = 0.03, SE = 0.01, Exp(B) = 1.03, *p* = 0.03. In particular, for each additional unit increase in sexual orientation, the frequency of risky sexting increases 1.19 times, while for each unit increase in FoMO, the frequency of risky sexting increases 1.03 times. See Table 1 for the statistics.

The negative binomial regression model for the second type of risky sexting (i.e., sexting for emotional regulation) accounted for 9.62% of the variance, R^2^ = 0.096, χ^2^(7) = 71.8, *p* < 0.001. Sexting for emotional regulation was positively associated with sexual orientation, B = 0.22, SE = 0.35, Exp(B) = 1.24, *p* < 0.001, and with total FoMO, B = 0.71, SE = 0.02, Exp(B) = 1.07, *p* < 0.001, while it showed a negative association with biological sex, B = −0.04, SE = 0.01, Exp(B) = 0.96, *p* = 0.005. Specifically, for each additional standard deviation of total FoMO, the frequency of sexting for emotion regulation increases 1.07 times, while a one-unit increase in sexual orientation leads to an increase of 1.24 times. Conversely, each additional standard deviation of biological sex is associated with a decrease of 1.04 times in sexting frequency.

Additionally, the interaction term between FoMO and sexual orientation was significant, B = 0.07, SE = 0.03, Exp(B) = 1.08, *p* = 0.03, indicating that for each unit increase in this interaction, the frequency of sexting for emotion regulation increases 1.08 times. See Table 1 for statistics.

To understand the direction of the interaction, a slope analysis was conducted by plotting sexting values as a function of FoMO for the two levels of sexual orientation (0 = exclusively heterosexual, 1 = not exclusively heterosexual). The slope analysis revealed a stronger positive and significant relationship for non-heterosexual individuals, B = 0.12, SE = 0.03, Exp(B) = 1.13, *p* < 0.001, compared to exclusively heterosexual individuals, B = 0.05, SE = 0.02, Exp(B) = 1.05, *p* = 0.03. Specifically, for non-heterosexual individuals, the frequency of sexting increases 1.13 times for each additional unit of the predictor, whereas for exclusively heterosexual individuals, the frequency increases 1.05 times for each additional unit. This indicates that the association between the predictor and sexting is stronger for non-heterosexual individuals (Figure 1).

## 4. Discussion

This study examined the connection between FoMO and three distinct sexting behaviors: experimental sexting (i.e., sharing of own sexts), and two kinds of risky sexting (i.e., sexting under substance use or with strangers and sexting for emotional regulation). To the best of our knowledge, it represents the first investigation into how FoMO directly can impact different kinds of sexting behaviors. Indeed, a previous study investigated the relationship between FoMO and sexting motivations ([44]) and others considered sexting behaviors, in general ([53]; [61]; [66]). Additionally, a study analyzed how age, biological sex, and sexual orientation are directly related to sexting and can moderate the relationship between FoMO and each type of sexting behavior, following the hypotheses tested by Morelli and colleagues (2024).

As regards the relationship between each socio-demographic variables and sexting, no significant effect of age (H2 not confirmed) was found for any of the three sexting behaviors analyzed, confirming the conflicting results present in the literature. Previous research on adolescents and young adults has shown that older individuals tend to engage in more sexting behaviors ([9]). The lack of an observed effect could be due to the homogeneity of the participants’ ages in the present study, as they all fell within the young adult category.

Regarding the effect of biological sex (H3 partially confirmed), a direct effect was found only for sexting for emotional regulation. In particular, the present study showed that females tend to engage more in sexting for emotional regulation, and this could be related to gender differences in emotional regulation strategies. Women are more likely to seek emotional support through social communication, while men may lean more towards individual coping strategies ([62]).

Sexual orientation emerged as having a direct effect on all three sexting behaviors: experimental sexting, sexting under the influence of substances or with strangers, and sexting for emotional regulation (H4 totally confirmed). The influence of sexual orientation on experimental sexting aligns with previous research, which has reported higher rates of sexting among LGB individuals compared to heterosexual individuals ([9]; [25]; [35]; [44]; [51]). Moreover, the fact that non-exclusively heterosexual individuals engage more frequently in risky sexual behaviors aligns with our expectations and reinforces previous findings, which indicate that sexual minorities are generally more prone to engaging in both high-risk sexual activities and risky sexting behaviors ([25]; [50]; [45]). Social media enhance communication and relationships while promoting psychological and socio-emotional well-being among lesbian, gay, and bisexual individuals ([14]). Consequently, they may serve as a protective factor against stressors related to social stigma, prejudice, and discrimination, as proposed by the Minority Stress Model ([40]), and can represent a space for greater sexual exploration.

As regards the relationship between FoMO and sexting, as expected (H1 partially confirmed), the findings of this study confirmed that FoMO is a significant predictor only of some kind of sexting behaviors. Indeed, no association was found between FoMO and experimental sexting. This result can be explained by the fact that experimental sexting is primarily driven by exploration and self-expression of one’s sexuality ([71]). Since it is not inherently considered a risky behavior, it may be less influenced by FoMO. On the other side, the results showed that FoMO has a direct influence on both types of risky sexting (i.e., sexting under substance or alcohol use or with strangers and sexting for emotional regulation). This finding can be understood within the framework of Self-Determination Theory ([17]), as stated by [54] ([54]). Individuals with elevated levels of FoMO often display a heightened drive for interpersonal connection coupled with reduced autonomy. As a result, those who frequently engage in sexting behaviors might be attempting to address unmet social and emotional needs and could be trying to fulfil their erotic and relational needs through chat-based interactions. This unmet need may sometimes drive individuals to engage in riskier sexual relational behaviors, as they may prioritize fulfilling these needs over considering the potential consequences of their actions. In such cases, sexting may become a high-risk behavior, driven by the urgency of addressing these emotional or social needs without fully assessing the risks involved.

In line with previous research ([44]), sexual orientation emerged as a moderator in the relationship between FoMO and the second type of risky sexting, i.e., sexting for emotional regulation. Therefore, this relationship was stronger and highly significant for non-heterosexual individuals, but weaker for heterosexual individuals. This finding suggests that FoMO has a stronger relationship with sexting among non-heterosexual individuals compared to heterosexual individuals. Specifically, for non-heterosexual individuals, higher levels of FoMO are associated with a greater likelihood of engaging in sexting for emotional regulation, potentially reflecting their reliance on digital platforms for social connection or to fulfil emotional needs, finding in online communication a tool for easier connection with others ([57]), along with the perception that it is safer than offline environments ([14]). In contrast, for heterosexual individuals, FoMO does not has a large influence on sexting behavior, indicating that other factors may play a more prominent role in driving these behaviors within this group. These results regarding the greater propensity of LGB individuals to engage in risky sexting behaviors can be understood considering the minority stress literature, which highlights a major propensity of those people to engage in risky activities, especially sexual behaviors ([56]).

## 5. Conclusions and Future Perspectives

This study highlights how certain variables that influence social media use can also play a role in determining sexual behaviors that occur through the internet, such as sexting. Studying FoMO and how it influences the exchange of explicit messages, photos, and videos via social networks and smartphones can help improve our understanding of the phenomenon and suggest directions for future research. Our results indicate that FoMO directly predicts only two types of risky sexting: sexting under substance use or with strangers, and sexting for emotional regulation, emphasizing how this factor could be a risk for problematic behaviors related to social media use ([29]). Additionally, our findings show that LGB individuals are more likely to engage in all types of sexting, and that FoMO is significantly related to engagement in risky sexting, especially for LGB young adults, highlighting a strong need for emotional fulfilment in this population.

However, this study has several limitations, which should be considered when interpreting its findings. First, its cross-sectional design limits the ability to establish causal relationships between the variables under investigation. Future studies should employ more complex statistical models to better assess the directional causality between FoMO and sexting behaviors, providing a deeper and more comprehensive understanding of these relationships. Another limitation to mention is the overrepresentation of women in the sample. Hopefully, future studies will achieve a better gender balance among participants.

Another limitation worth mentioning pertains to the relatively smaller representation of LGB individuals within the sample. Although the proportion aligns with that observed in the general population, future studies should aim to include a larger number of LGB participants. This would enable more robust analyses across subgroups of sexual orientation, providing deeper insights into the unique dynamics and experiences of these populations.

Finally, one last limitation to mention is the low explained variance of the model tested in the present study, which can be understood in light of the complex and multifactorial nature of the behavior being investigated. It is likely that other relevant variables, not accounted for in the present study, contribute to the phenomenon and further explain its variance.

A future direction for upcoming studies involves examining the same variables within an adolescent sample. This approach could help uncover potential differences in dynamics compared to young adults and inform the development of tailored interventions designed specifically to address the unique needs and challenges faced by adolescents.

However, these findings have several implications for prevention programs. It is crucial to implement targeted interventions aimed at digital education and the prevention of sexting-related risks, not only among adolescents but also young adults, who are often overlooked in such projects. These programs should prioritize fostering a sense of autonomy, addressing the developmental needs of growth and individuation, and supporting young people in navigating the challenges of online interactions. Furthermore, interventions should focus on enhancing body image self-esteem and promoting mutual respect in digital communication, with particular attention to individuals from marginalized communities who may face unique vulnerabilities. Prevention efforts should focus on providing young people the skills and information needed to engage in sexting responsibly. This includes raising awareness about the potential risks and consequences of their actions while fostering individual and collective resources to promote a safer online environment.

## Figures and Tables

**Figure 1 behavsci-15-00454-f001:**
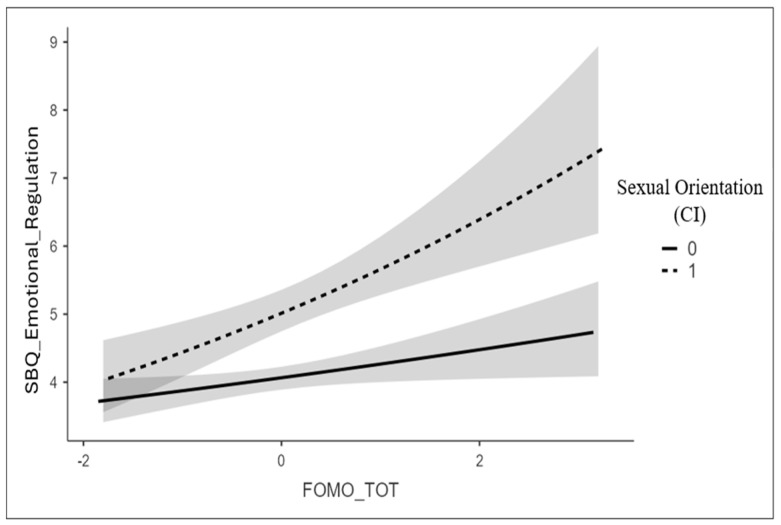
The moderating role of sexual orientation in the relationship between FoMO and sexting for emotion regulation. Note: Slope analysis for sexting for emotional regulation and FoMO, as a function of sexual orientation (0 = exclusively heterosexual; 1 = not-exclusively heterosexual).

**Table 1 behavsci-15-00454-t001:** Negative binomial regression analysis for each sexting dimension.

	Experimental Sexting						Sexting Under Substance Use						Sexting for Emotional Regulation					
Predictors	B	SE	Exp(B)	LCI	UCI	*p*	B	SE	Exp(B)	LCI	UCI	*p*	B	SE	Exp(B)	LCI	UCI	*p*
A	−0.01	0.02	1.00	0.96	1.02	0.48	0.01	0.01	1.01	0.98	1.04	0.39	0.03	0.02	1.03	1.00	1.06	0.08
BS	−0.02	0.02	0.98	0.95	0.01	0.26	−0.02	0.01	0.98	0.95	1.01	0.17	**−0.04**	**0.02**	**0.96**	**0.93**	**0.99**	**0.00**
SO	**0.16**	**0.03**	**1.17**	**1.10**	**1.26**	**<0.001**	**0.17**	**0.03**	**1.19**	**1.12**	**1.27**	**<0.001**	**0.22**	**0.03**	**1.24**	**1.16**	**1.33**	**<0.001**
FoMO	0.01	0.02	1.01	0.97	1.04	0.67	**0.03**	**0.02**	**1.03**	**1.00**	**1.07**	**0.03**	**0.07**	**0.02**	**1.07**	**1.04**	**1.11**	**<0.001**
FoMO × A	0.01	0.02	1.01	0.98	1.05	0.51	0.01	0.01	1.01	0.98	1.04	0.37	0.00	0.02	1.00	0.97	1.03	0.93
FoMO × BS	0.01	0.02	1.01	0.97	1.04	0.61	−0.004	0.01	1.00	0.97	1.03	0.77	−0.02	0.02	0.98	0.94	1.01	0.14
FoMO × SO	0.01	0.03	1.01	0.95	1.09	0.67	0.03	0.03	1.03	0.97	1.09	0.34	**0.07**	**0.03**	**1.08**	**1.01**	**1.15**	**0.03**

Note: Biological sex was coded as 0 = men; 1 = women; sexual orientation was coded as 0 = heterosexual participants; 1 = not-exclusively-heterosexual; FoMO = fear of missing out; A = age; BS = biological sex; SO = sexual orientation; LCI = low confidence interval; UCI = upper confidence interval. The significant results are highlighted in bold.

## Data Availability

The data presented in this study are available on request from the corresponding author due to ethical reasons.

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
