# Peer review of "Sexting Behaviors and Fear of Missing out Among Young Adults"

_behavsci, 2025, doi:10.3390/bs15040454_

Round 1
Reviewer 1 Report
Comments and Suggestions for Authors
The theoretical framework provides a comprehensive description of FOMO and its associated psychological effects and phenomena. The article also addresses the issue of sexting, differentiating its various types and identifying associated risks, along with age and gender specificities. The review offers a good overview of the current state of research in these areas. The authors convincingly demonstrate a gap in the research on the relationship between FOMO and sexting, and propose self-determination theory as a suitable methodological framework for investigating this link. The hypotheses posit the predictive power of FOMO in different types of sexting, as well as the contribution of age, gender, and sexual orientation to the FOMO-sexting relationship. The sample size comprised 911 participants. The methods employed are appropriate for testing the stated hypotheses. The results are presented clearly and informatively. The discussion section justifies the novelty of the findings, specifically addressing the direct effects of FOMO on various sexting behaviors. The results are well-interpreted and contextualized within existing research. The conclusion outlines future research directions and limitations. An additional limitation, concerning the significant overrepresentation of women in the sample, could be added. Overall, the work is of high quality and meets the main requirements.
Author Response
The reviewer 1 wrote:
“An additional limitation, concerning the significant overrepresentation of women in the sample, could be added”
RESPONSE: Thank you for this comment, which allows us to clarify our work. The following sentence has been added to the text in the Conclusions and Future Perspectives section as follows: “Another limitation to mention is the overrepresentation of women in the sample. Hopefully, future studies will achieve a better gender balance among participants”.
Reviewer 2 Report
Comments and Suggestions for Authors
Excellent paper succinctly written with clear methodological insights making it relatively easy to replicate in other jurisdictions, and with younger adolescents, in terms of the measures used, reasons this area is important and theoretical understanding of FoMO and sexting behaviours.
Literature review was broad and up to date as well as including some seminal works, clearly indicated why the study had merit.
Only criticism - it was not clear how they accessed their sample. Who were the young adults? how was the study advertised/sold to participants? - was there bias then in terms of those who choose to fill out the survey?

Author Response
We appreciate the reviewer’s positive feedback and insightful comments. We have made the requested clarification. Below, we provide a point-by-point response to each suggestion.
The reviewer 2 wrote: Only criticism - it was not clear how they accessed their sample. Who were the young adults? how was the study advertised/sold to participants? - was there bias then in terms of those who choose to fill out the survey?
RESPONSE: Thank you for this comment, which allows us to clarify our methodology.
The following sentene was added in the participants and procedure section:
“Young adults’ participants were recruited from various Italian universities through the distribution of an anonymous link containing the research questionnaires. Partici-pants were also asked to share the link to the survey among their contacts and on their social network’s platforms. This facilitated a snowball sampling approach as the survey was further shared online. The study was presented to participants as research on re-lationships in the digital context, and participation was entirely voluntary. A potential bias stems from the voluntary nature of participation and the non-probabilistic sam-pling method, which is common in most psychological studies.”
Reviewer 3 Report
Comments and Suggestions for Authors
First of all, thank you for giving me the opportunity to review this article.
I would divide the introduction into sub-paragraphs to help the reader in reading.
I would explain the research hypotheses well, taking them up in the discussions.
In general the variance explained is not large, a section could be included where the question of possible reasons is discussed.
Apart from these minor revisions the article is certainly interesting and deserves to be published
Author Response
We are grateful for the reviewer's thoughtful feedback and constructive comments. We have addressed the requested clarifications and we provide a point-by-point response below.
- The reviewer 3 wrote:
“I would divide the introduction into sub-paragraphs to help the reader in reading”.
RESPONSE: We thank the reviewer for this insightful comment, which allows us to better structure our article. The introduction has been divided into paragraphs for improved readability.
- The reviewer 3 wrote:
“I would explain the research hypotheses well, taking them up in the discussions.”
RESPONSE: Thank you for bringing this point to our attention. We have tried to simplify the research hypotheses section by presenting them as bullet points, and we now take them in the discussion section.
- The reviewer 3 wrote:
“In general the variance explained is not large, a section could be included where the question of possible reasons is discussed?
RESPONSE: We thank the reviewer for this comment. The relatively small proportion of explained variance in this study can be attributed to the complex and multifactorial nature of sexting behavior. It is likely that other relevant variables, not accounted for in the present study, contribute to the phenomenon and further explain its variance. Additionally, in psychological research, it is typical for the explained variance to be modest, as behaviors are often influenced by a multitude of interacting factors. Furthermore, recent studies in this field have reported similar levels of explained variance, reinforcing the validity of these findings (ad es. Morelli et al., 2025).
A paragraph in the limits section has been included as follows: “Finally, one last limitation to mention is the low explained variance of the model tested in the present study, which can be understood in light of the complex and mul-tifactorial nature of the behavior being investigated. It is likely that other relevant variables, not accounted for in the present study, contribute to the phenomenon and further explain its variance.